# Assessment of Health-Related Quality of Life in Patients with Advanced Prostate Cancer—Current State and Future Perspectives

**DOI:** 10.3390/cancers14010147

**Published:** 2021-12-29

**Authors:** Alexander Kretschmer, Roderick C. N. van den Bergh, Alberto Martini, Giancarlo Marra, Massimo Valerio, Igor Tsaur, Isabel Heidegger, Veeru Kasivisvanathan, Claudia Kesch, Felix Preisser, Christian D. Fankhauser, Fabio Zattoni, Francesco Ceci, Jonathan Olivier, Peter Chiu, Ignacio Puche-Sanz, Constance Thibault, Giorgio Gandaglia, Derya Tilki

**Affiliations:** 1Department of Urology, Ludwig-Maximilians University, 81377 Munich, Germany; 2Department of Urology, Antonius Hospital, 3543 AZ Utrecht, The Netherlands; roodvdb@hotmail.com; 3Unit of Oncology/Unit of Urology, Urological Research Institute, IRCCS Ospedale San Raffaele, 20132 Milan, Italy; alberto.martini.90@gmail.com (A.M.); giorgio.gandaglia@gmail.com (G.G.); 4Department of Urology, San Giovanni Battista Hospital, University of Torino, 10126 Torino, Italy; marragiancarlo@hotmail.it; 5Department of Urology, CHUV Lausanne, 1011 Lausanne, Switzerland; 7massimo7@gmail.com; 6Department of Urology and Pediatric Urology, Mainz University Medicine, 55131 Mainz, Germany; Prof.Dr.med.Igor.Tsaur@unimedizin-mainz.de; 7Department of Urology, Medical University Innsbruck, 6020 Innsbruck, Austria; isabel.heidegger@tirol-kliniken.at; 8Division of Surgery and Interventional Science, University College London, London W1W 7TY, UK; veeru.kasi@ucl.ac.uk; 9Department of Urology, University Hospital Essen, 45147 Essen, Germany; Claudia.Kesch@uk-essen.de; 10Department of Urology, University Hospital Frankfurt, 60590 Frankfurt, Germany; felixpreisser@gmx.de; 11Luzerner Kantonsspital, 6000 Lucerne, Switzerland; cdfankhauser@gmail.com; 12Department of Surgery, Oncology and Gastroenterology, University of Padua, 35128 Padua, Italy; fabiozattoni@gmail.com; 13Urology Unit, Academical Medical Centre Hospital, 33100 Udine, Italy; 14Division of Nuclear Medicine, IEO European Institute of Oncology, 20141 Milan, Italy; francesco.ceci83@gmail.com; 15Department of Urology, Lille University Hospital, 59000 Lille, France; jo_olivier@msn.com; 16SH Ho Urology Centre, Department of Surgery, The Chinese University of Hong Kong, Hong Kong, China; peterchiu@surgery.cuhk.edu.hk; 17Department of Urology, Bio-Health Research Institute, Hospital Universitario Virgen de las Nieves, University of Granada, 18014 Granada, Spain; nacho.puchesanz@gmail.com; 18Department of Medical Oncology, European Georges Pompidou Hospital, Assistance Publique des Hôpitaux de Paris, Paris Descartes University, 75004 Paris, France; constance.thibault@egp.aphp.fr; 19Martini-Klinik Prostate Cancer Center, University Hospital Hamburg-Eppendorf, 20246 Hamburg, Germany; dtilki@me.com; 20Department of Urology, University Hospital Hamburg-Eppendorf, 20246 Hamburg, Germany; 21Department of Urology, Koc University Hospital, 34010 Istanbul, Turkey

**Keywords:** prostate cancer, metastatic, health-related quality of life, QLQ-C30, EQ-5D, FACT-P

## Abstract

**Simple Summary:**

In recent years, evidence regarding survival outcomes of novel therapies has increased significantly. However, less is known regarding the impact of novel therapy combinations on quality of life aspects of the treated patients. In the current comprehensive review, we discuss the importance of quality of life for patients with advanced prostate cancer, and present the most frequently used tools to evaluate quality of life in recent randomized trials. Further, we discuss the ease of use of these validated questionnaires for clinicians and try to focus on the suggested appropriate use as well as potential strategies for improvement of quality of life evaluation in these clinical scenarios of advanced prostate cancer.

**Abstract:**

With the therapeutic landscape of advanced prostate cancer rapidly evolving and oncological benefits being shown for a plethora of new agents and indications, health-related quality of life (HRQOL)-associated evidence is still subpar. In the current comprehensive review, we discuss the importance of HRQOL for patients with advanced PC (metastatic hormone-sensitive prostate cancer (mHSPC), metastatic castration-resistant prostate cancer (mCRPC) and non-metastatic castration-resistant prostate cancer (nmCRPC)), and present the most frequently used tools to evaluate HRQOL in recent randomized trials. Furthermore, we discuss the ease of use of these validated questionnaires for clinicians and try to focus on the suggested appropriate use in clinical practice, as well as potential strategies for improvement of HRQOL evaluation in these clinical scenarios of advanced prostate cancer.

## 1. Introduction

“I want to live better, longer and maintain quality of life” is a hypothetical patient’s quote that almost every treating physician might already have heard when discussing treatment strategies with patients with advanced prostate cancer (PC). While the “longer” part has been adequately addressed in recent years, widely accepted endpoints such as overall survival and cancer specific survival can be used to integrate respective results and findings in the greater picture of the currently available plethora of treatment options to the individual patient in the advanced PC setting. When it comes to maintaining quality of life, less evidence is available and a lack of validated and universally acknowledged health-related quality of life (HRQOL) endpoints as well as use of different tools make it challenging to interpret currently available data. However, it has to be acknowledged that in recent years, evidence regarding HRQOL in the advanced PC setting has significantly improved, and federal authorities have put more focus on HRQOL outcomes during the approval process. Clinicians and researchers have become more used to assessing patient-reported outcome measures (PROMs), which are most frequently evaluated through validated multidimensional questionnaires. 

In the current comprehensive review, we discuss the importance of HRQOL for patients with advanced PC (metastatic hormone-sensitive prostate cancer (mHSPC), metastatic castration-resistant prostate cancer (mCRPC) and non-metastatic castration-resistant prostate cancer (nmCRPC)), and present the most frequently used tools to evaluate HRQOL in recent randomized trials. Further, we discuss the ease of use of these validated questionnaires for clinicians and try to focus on the suggested appropriate use as well as potential strategies for improvement of HRQOL evaluation in these clinical scenarios of advanced PC.

## 2. Importance of HRQOL for Patients with Advanced Prostate Cancer

### 2.1. Availability of Novel Therapies

In 2004, SWOG 99-16 [1] and TAX-327 [2] presented the advantage of docetaxel chemotherapy treatment for patients with mCRPC. Before that time, no therapies with a substantial survival benefit over hormonal therapy were available for patients with advanced prostate cancer (PC). Since then, the arsenal of treatment options found to have superior results when added to standard hormonal therapy in randomized controlled trials has considerably expanded. The indication for these new therapies has also shifted from mCRPC to mHSPC and to nmCRPC. Options now available in addition to standard androgen deprivation therapy (ADT) include taxane-based chemotherapy [3,4,5,6], new androgen-receptor targeted agents (ARTAs; abiraterone, enzalutamide and more recently apalutamide and darolutamide) and other options (Radium-223 [7], Sipuleucel-T [8], local radiotherapy of the prostate [9], Lu-177 PSMA [10], Olaparib [11]). The arena for patients with advanced PC is rapidly evolving. 

### 2.2. Side Effects of Novel Therapies

The different new available therapies have a different spectrum of potential side effects that may affect HRQOL. Serious side effects caused by chemotherapeutical agents mainly relate to bone marrow depression, such as neutropenia, thrombocytopenia and anemia. In addition, fatigue, neuropathy and gastrointestinal symptoms have been described. With the use of ARTAs, side effects specific for their method of action may occur. Abiraterone may cause hypertension, edema or electrolyte imbalance due its impact on adrenal gland function. Cardiac events may also occur. Enzalutamide may cause fatigue, hypertension, hot flushes and neurological symptoms such as seizures.

### 2.3. Benefit of Novel Therapies

Most RCTs investigating the effect of novel therapies have used overall survival (OS) as the main endpoint for sample size power calculations. Other endpoints include cancer-specific survival and metastatic progression. The benefit of all new therapies is dependent on tumor characteristics and, although many uncertainties remain, patients with higher risk disease and metastatic load seem to have a higher benefit of an early start of these agents in addition to hormonal therapy only. In addition, the timing and sequencing of all available treatments is not yet fully understood. Patients with a durable response on hormonal manipulation however, may be more likely to respond well to ARTAs in first line, while patients rapidly progressing after ARTAs in second line may have more benefit from chemotherapy as the third therapy line [12]. 

### 2.4. Balancing Benefits and Risk of Adverse Events

While survival is unquestionably a relevant outcome, less is known on the impact of therapy choice on HRQOL of a patient. HRQOL may be positively impacted by response to a treatment causing improvement of clinical symptoms, but may on the other hand be negatively impacted due to side effects. In the treatment decision process in localized PC, there are no significant differences between treatments in oncological outcomes, but a clear distinction in patterns of side effects with potential impact on HRQOL [13]. Compared to this clinical situation, HRQOL issues in patients with advanced PC are a relatively understudied field in the literature. Direct comparisons in HRQOL outcomes between different treatment options are scarce. While for a physician, oncological endpoints (“live longer”) may dominate in deciding on timing and choice of specific therapy, HRQOL life issues may actually be more important from a patient’s perspective (“maintain quality of life”). It is also very important to take patient specific factors such as comorbidity and age into account. Especially for patients with mCRPC who have to decide on second, third, fourth or even fifth lines of treatment the balance between life expectancy on one side and the HRQOL on the other is essential. These men frequently have poor performance status, with pain due to bone metastasis dictating their life. 

### 2.5. Measuring Health-Related Quality of Life (HRQOL)

HRQOL for a patient with advanced prostate cancer is composed of specific urological-related symptoms, and more general overall HRQOL. The latter may be measured using different tools. The sensitivity of a questionnaire for change in HRQOL depends on different aspects. 

First, the disease setting, when is the questionnaire applied; while for a patient with localized PC it is unlikely that treatment choice will impact a measure such as “are you able to climb a stairs”, it is likely that this outcome is relevant for a patient with advanced PC deciding on treatment choice in fourth line systemic therapy. In line with this, urological symptoms such as erectile dysfunction or incontinence may be pushed to the background when a patient enters the advanced disease stage. 

Second, expected impact, what does the questionnaire measure; questionnaires applied in men with metastatic PC should ideally cover issues likely to occur or improve due to therapies; such as fatigue, pain, performance or the impact of side effects. The more specific measures relate to issues that can be expected to arise during treatment, the more likely it is that any changes will be detected. The benefit of therapy in terms of oncological outcomes and improved quality of life due to response should be balanced with the potential negative impact due to side effects, burden of therapy and risk of adverse events.

## 3. Currently Used Validated Questionnaires

### 3.1. EQ-5D

The European Quality of Life 5-Dimensions questionnaire (EQ-5D) was introduced in the 1990s as a result of a multi-institutional effort by the European Quality of Life Group that began in the late 1980s [14]. The research group, composed by researchers from Northern Europe, aimed to produce an instrument that was not disease-specific, but could rather be applied in different contexts. Over the past decades, this tool has been used in population health surveys, clinical studies and in economical evaluation studies, making it a very versatile questionnaire [15]. 

The EQ-5D has been designed for self-completion (i.e., by the individual that is subject of the study). It has two main components: the health state description and the evaluation. The first part evaluates five dimensions (5D): Mobility, self-care, usual activities, pain/discomfort and anxiety/depression. Mobility dimension refers to individual’s walking ability. The self-care dimension regards the ability to wash or dress by oneself. The usual activities dimension measures performances in work, study, housework, family or leisure activities. The last two domains of the EQ-5D enquire about how much pain/discomfort one has and how much anxiety/depression the individual perceives. The second component of the EQ-5D consists of a visual analogue scale (VAS) where the individual is asked to evaluate his/her overall health status. The scale spans from 0 to 100 where 0 corresponds to the worst health status one can imagine and 100 to the best possible one. 

Three different versions of the EQ-5D exist, the “youth version” [16], the “3-level” (3L) [14] and the “5-level” 5L [17] and are available on the European Quality of Life website (https://euroqol.org; accessed on 25 May 2021). The 3L and 5L versions have been employed in studies on advanced prostate cancer. The EQ-5D-3L has three possible levels for each of the five domains: No problems, some problems and extreme problems, whereas the EQ-5D-5L: no problems, slight problems, moderate problems, severe problems and being unable to do/having extreme problems. On both, the VAS scale is unchanged.

The EQ-5D has a moderate use in the current trial landscape for advanced prostate cancer [18]. Out of 14 trials evaluating novel treatments in the context of advanced prostate cancer, only six adopted the EQ-5D in an effort to evaluate patient’s quality of life [18].

### 3.2. EORTC QLQ-C30

The European Organization for Research and Treatment of Cancer Quality of Life Questionnaire (EORTC QLQ-C30) is a patient-reported outcome “core questionnaire” first developed in the late eighties and released in 1993 [19,20]. The EORTC goal was to allow HRQOL evaluation in a cancer-specific manner, with an easy self-administration and applicable through multiple socioeconomic and cultural settings. Four different versions have been released with Version 3.0 being the most recent. 

QLQ-C30 is a generic instrument to allow a common and standardized HRQOL assessment throughout all cancer patients. To obtain HRQOL specificity depending on the type of cancer the core questionnaire (QLQ-C30) has to be combined with cancer-specific questionnaires defined as “modules” which may include cancer-specific symptoms (e.g., urinary for PC) and/or treatment related side effects (e.g., surgery, chemotherapy).

The modular approach adopted by the EORTC allows both generic and specific HRQOL measurements to compare different subgroups depending on cancer-related treatment and stage but also amongst different cancer groups. Currently, more than twenty modules have been validated including prostate, colorectal, breast and other cancers whilst more than thirty others are under development/validation. The Prostate Cancer Module to be used in conjunction with the QLQ-C30 is the EORTC QLQ-PR25 [21].

The core QLQ-C30 questionnaire consists of 30 questions. Answers are based on a four-point (n = 28 items—1 = “Not at all” to 4 = “Very Much”) or seven-point scale (n = 2 items—1 = “Very poor” to 7 = “Excellent”) with the majority of items referring to the time period within the previous week. For final assessment, questionnaire values have to be converted in a 0 to 100 linear scale with higher scores indicating better functioning for the functioning and global HRQOL domains but worse symptoms for the symptom domains. Overall, QLQ-C30 questions comprise five different function domains (physical, social, emotional, role and cognitive) and evaluate eight symptoms (fatigue, pain, insomnia, dyspnea, constipation, diarrhea, nausea and/or vomiting and appetite loss). Financial impact is also considered. 

The QLQ-C30 has been validated in several studies including assessment of its psychometric properties, ability to differentiate patients with distinct performance statuses and receiving different treatments [22], and has high patient–observer ratings agreement [23]. Core instrument and related modules so as for their translations and deliverability in trials have all been developed and validated according to specific guidelines [24,25]. All documents and regular updates are on the EORTC website and free of charge if not to be used in sponsored studies (https://qol.eortc.org; accessed on 25 May 2021).

Currently, EORTC QLQ-C30 remains one of the most widely used instruments to evaluate HRQOL in cancer patients. More than 50 translations and local adaptations have been officially released to expand its availability not only in Western Countries but also to Africa, Asia and South America.

From 2004 to 2019, almost one out of three of the *n* = 120 studies reporting HRQOL from PC randomized-controlled trials used QLQ-C30 and/or its related PR25 module. This makes the QLQ-C30 one of the most frequently used HRQOL measure for PC together [26]. Conversely, in recent trials evaluating HRQOL in advanced PC, the use of EORTC-based questionnaires was less consistent (3 of 14 RCTs) [18].

### 3.3. FACT-P

The Functional Assessment of Cancer Therapy Prostate (FACT-P) questionnaire has been developed in the early 1990s aiming at providing clinicians a multidimensional, user-friendly and versatile tool to assess HRQOL in men with localized and advanced PC [27]. This effort mirrored what has been carried out for other malignancies ranging from GI to gynecological cancers. In addition to a general questionnaire, named FACT-General (FACT-G), the FACT-P was developed to capture disease-specific domains [28]. The modern FACT-P is represented by a comprehensive questionnaire including 27 cancer-specific items (FACT-G) plus 12 PC-specific items. The 27 cancer-specific items encompass four domains: physical well-being (seven items), social and family well-being (seven items), emotional well-being (six items) and functional well-being (seven items). The 12-items PC-specific part explores general symptoms, genitourinary and rectal functional status. Every question has a five Likert type scale ranging from 0 = “not at all”–to 4 = “very much”. The combination of the scale provides a global HRQOL score as well as domain-specific scores. 

The modern FACT-P version 4.0 is a solid and powerful PROM (patient-reported outcome measurement). It has been validated in men with localized and advanced disease and it seems to be versatile to many men with variable socioeconomic background. From a researcher as well as patient’s perspective, it is quick as it requires around 8 to 10 min to be completed within a trial setting and it has been shown to have good internal consistency and concurrent validity for PC disease. Of key importance, unlike other more general PROM, it is sensitive to change in performance status, disease migration as well as PSA change. In other words, the variation between baseline and post-treatment FACT-P informs clinicians about the oncological and the general well-being all in one in many cases.

In light of the aforementioned characteristics, the FACT-P is the most commonly used PROM to assess HRQOL in men with advanced disease participating in confirmatory clinical trials. According to a recent systematic review, around 80% (11/14) encompassed the FACT-P in phase 3 trials evaluating novel agents in advanced disease in the last decade [18]. Regardless the target population investigated—as either mHSPC, nmCRPC or mCRPC—FACT-P proved to be useful in determining the functional status, and importantly in most cases its variation reflected the response to treatment, as aimed in the development phase of the tool [18]. 

## 4. Ease of Use for Clinicians

### 4.1. Appropriate Use in Current Trials

Table 1 summarizes HRQOL outcomes from recent RCTs that focused on advanced PC. Detailed results of the HRQOL outcomes of the respective trials have recently systematically reviewed by our working group [18]. 

Since there are multiple validated questionnaires available that are frequently used for evaluation of HRQOL in clinical trials involving advanced prostate cancer patients, it is a challenging task to integrate this data into current therapy regimens. One potential pitfall encompasses the fact that, as shown above, even though all of the most frequently used questionnaires evaluate HRQOL in a multidimensional and validated fashion, the respective items and domains vary significantly and comparison of the results is not trivial. Using sophisticated statistical methods in order to map the results of FACT- and EQ-5D-based utility scores in cancer patients, Pickard et al. found mixed results for different approaches and were not able to provide clear recommendations [41]. Similar studies have been performed in patients with mCRPC to provide guidance for mapping the FACT-P to the EQ-5D questionnaire for use in cost-effectiveness analyses [42] or to enable calculation of EQ-5D scores when EQ-5D data have not been assessed directly [43]. However, to date there is still no generally accepted guideline accessible that enables the robust mapping of different questionnaires. To address this issue, more recent RCTs implemented several concomitant questionnaires. For instance, the recently published LATITUDE [29], TITAN [31] and SPARTAN [35] trial included the FACT-P as well as the EQ-5D questionnaire in their assessment. The investigators of the PROSPER [33] and ARCHES [32] went even further and even added the QLQ-PR25 questionnaire to their arsenal. Recently, assessment of HRQOL in ARAMIS trial was published [34]. Hereby, HRQOL was assessed using the FACT-P as well as the QLQ-PR25 questionnaire. Darolutamide maintained significantly delayed time to deterioration of prostate cancer–specific HRQOL and disease-related symptoms versus placebo.

Another indicator for appropriate use in current trials is the reporting of baseline values. As indicated in Table 1, reporting of baseline values has more and more evolved as a quality standard of recent RCTs but has not been performed in all the mentioned studies. It is generally accepted that pretreatment HRQOL is an important predictor of HRQOL during the respective treatment course. In addition, reporting of baseline values allows for better indirect comparison between the respective studies (Figure 1) and therefore improves contextualization of the data, e.g., for hypothesis-generating purposes.

Another pitfall of the scores that are generated by the different HRQOL measurement tools is the ongoing debate if, and how, these changes actually display clinical meaningfulness. Again, these definitions vary between recent studies and there is still no generally accepted consensus. Still, investigators should define clinical meaningfulness in advance, describe the definition in the materials and methods section and report the results accordingly. Future efforts should encompass general recommendations for definitions of clinical meaningfulness in order to further improve contextualizing of the results and to improve the potential of HRQOL-associated endpoints. In this context, it has to be emphasized that recently, cut-off values for the QLQ-C30 questionnaire have been published, allowing for valid stratification for good general HRQOL based on the QLQ-C30 global health status domain [44]. Thus, dichotomizing patients into subgroups with and without “good general HRQOL” based on the results of PROM assessment has become more feasible.

### 4.2. Patient Uptake, Potential Strategies for Improvements

Patient uptake and missing data is a broadly accepted issue in questionnaire-based science. Regarding recent RCTs in advanced prostate cancer, reported missing data rates varied significantly between the respective studies, e.g., 33% for the CHAARTED trial [30] and 10% in the LATITUDE trial [29]. In addition, there is an obvious trend towards higher missing-data rates in the longer-term follow-up.

Nowadays, internet-based HRQOL measurement tools that can also be used on mobile devices can offer a potential patient uptake benefit. While there are still some country-specific legal questions to answer, these digital options have already been implemented in daily clinical practice. For instance, a mobile app version of the QLQ-30 is also available and has been validated showing high versatility with approximately four minutes needed to fill out the questionnaire. The majority of patients also preferred the online over the paper version and stated they would support its use [45]. Another example is the international multi-institutional Prostate Cancer Outcome (PCO) study that is partially funded by the Movember foundation and uses online tools in order to assess the EPIC-26 questionnaire following definitive therapy of localized PC. More and more departments integrate patient-reported outcomes in electronic patient files and questionnaires can be completed by using online patient portals. However, if these scores are integrated in electronic patient files, it can be debated whether subdomain scores or overall scores should be reported. In addition, while usage of validated questionnaires should be promoted, health-care providers must also make sure to give patients the opportunity to highlight specific questions that are important to them. Further pitfalls of clinical application encompass visualization of the outcomes for the respective patients as well as assessment longitudinal measurements. Finally, current trials have different follow-up times in each study arm which might lead to potential under-reporting of long-term adverse events and consecutively impact HRQOL outcomes [46].

## 5. Conclusions and Future Perspectives

With the therapeutic landscape of advanced prostate cancer rapidly evolving and oncological benefits being clearly shown for a plethora of new agents and indications, HRQOL-associated evidence is also improving but still subpar. While it has become standard of care to report HRQOL outcomes in RCTs, differences in the methodology still hamper the comparability of the data and efforts should be undertaken to address these issues and to further highlight the potential of HRQOL as a primary endpoint. Future studies should aim to present direct comparisons of HRQOL outcomes of different available agents in order to facilitate these aspects in clinical decision making.

## Figures and Tables

**Figure 1 cancers-14-00147-f001:**
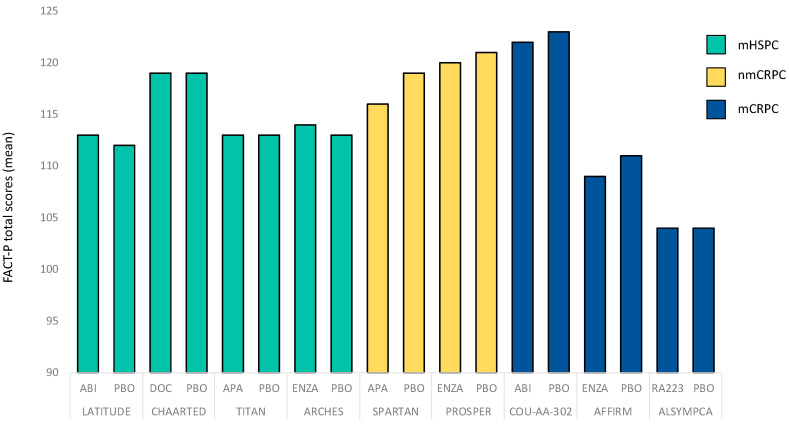
Baseline FACT-P total scores of selected contemporary randomized-controlled trials that focused on metastatic hormone-sensitive prostate cancer (mHSPC), nonmetastatic castration-resistant prostate cancer (nmCRPC) or metastatic castration-resistant prostate cancer (mCRPC). Hereby, higher FACT-P total scores correlate with increased well-being.

**Table 1 cancers-14-00147-t001:** Summary of selected contemporary randomized-controlled trials investigating health-related quality of life in patients with advanced prostate cancer (ABI = abiraterone acetate; ADT = androgen deprivation therapy; APA = apalutamide; CAB = cabazitaxel; DOC = docetaxel; ENZA = enzalutamide; HRQOL = health-related quality of life; mCRPC = metastatic castration-resistant prostate cancer; mHSPC = metastatic hormone-sensitive prostate cancer; nmCRPC = nonmetastatic castration-resistant prostate cancer; RA223 = radium-223 dichloride).

Study	Year	Clinical Stage	Intervention	HRQOL Primary Endpoint	HRQOL Assessment Tool	HRQOL Baseline Values Reported
LATITUDE [29]	2018	mHSPC	ABI vs. PBO	no	FACT-P EQ-5D (-5L)	yes
E3805 CHAARTED [30]	2018	mHSPC	DOC + ADT vs. ADT	no	FACT-P (FACT-Taxane)	yes
TITAN [31]	2019	mHSPC	APA vs. PBO	no	FACT-P EQ-5D (-5L)	yes
ARCHES [32]	2020	mHSPC	ENZA vs. PBO	no	FACT-P EQ-5D (-5L) QLQ-PR25	yes
PROSPER [33]	2019	nmCRPC	ENZA vs. PBO	no	FACT-P QLQ-PR25 EQ-5D (-5L)	yes
ARAMIS [34]	2021	nmCRPC	DARO vs. PBO	no	FACT-P QLQ-PR25	
SPARTAN [35]	2018	nmCRPC	APA vs. PBO	no	FACT-P 5Q-5D (-3L)	yes
PREVAIL [36]	2017	mCRPC	ENZA vs. PBO	no	EQ-5D (-3L)	yes
AFFIRM [37]	2014	mCRPC	ENZA vs. PBO	no	FACT-P	yes
ALSYMPCA [7]	2016	mCRPC	RA223 vs. PBO	no	FACT-P EQ-5D (-5L)	yes
PROSELICA [5]	2017	mCRPC	CAB 20 vs. CAB 25	no	FACT-P	no
FIRSTANA [38]	2017	mCRPC	CAB 20 vs. CAB 25 vs. DOC	no	FACT-P	no
COU-AA-301 [39]	2011	mCRPC	ABI + ADT vs. PBO + ADT	no	FACT-P	no
COU-AA-302 [40]	2013	mCRPC	ABI + ADT vs. PBO + ADT	no	FACT-P	yes

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
