# Peer review of "Assessment of Health-Related Quality of Life in Patients with Advanced Prostate Cancer—Current State and Future Perspectives"

_cancers, 2021, doi:10.3390/cancers14010147_

Round 1

Reviewer 1 Report

The review is generally well-written and informative.

Few minor comments.

Is it YAU Prostate Cancer Working “Party” or “Group”?

Page 3, under 2.3. Benefit of novel therapies: the sentence: “In the mCRPC setting, the absolute benefit in OS versus ADT alone….” is unclear. What are the authors saying here?

Under 2.4. instead of underexposed field, one may write understudied field

Page 5, PROM. What is the meaning of “PROM”?

Sentence: reporting of baseline values has been more and more evolved as a quality standard of recent RCTs but has not been performed in all the displayed studies. Perhaps: reporting of baseline values has more and more evolved as a quality standard of recent RCTs but has not been performed in all the mentioned studies.

Figure 1: FACT total scores – y-axis. What is the meaning of the shown scores? Should be explained.

Page 8, first paragraph, sentence: In this context, it has to be emphasized…The sentence is unclear. What is the meaning of the cut-off values?

Same page, next paragraph, last sentence: “Finally….” What is over-reporting of adverse events? It would imply that some adverse events are reported that did not occur.

Reviewer 2 Report

Title: Assessment of health-related quality of life in patients with advanced prostate cancer – current state and future perspectives  

The review is an interesting landascape concerning health related quality of life in patients with advanced prostatic cancer. It is a very current item.

Authors made an overview of the more appropriate questionnairs before pointed out some clinical criticisms about their use.

The paper is easily readable.

Author Response

Thank you very much for your encouraging comment.
